

# Effects of aqueous and ethanolic extracts of Chinese propolis on dental pulp stem cell viability, migration and cytokine expression

Ha Bin Park, Yen Dinh, Pilar Yesares Rubi, Jennifer L. Gibbs and Benoit Michot

Department of Restorative Dentistry and Biomaterials Sciences, Harvard School of Dental Medicine, Boston, MA, United States

## ABSTRACT

**Background:** Propolis is a natural substance produced by honeybees that has various biological properties including, anti-inflammatory, antioxidant and antimicrobial properties. Although previous studies have evaluated the antimicrobial effects of propolis in dentistry, its effects on dental pulp stem cell (DPSC) viability, migration, and differentiation are yet not well understood. The objective of this study was to investigate the effects of Chinese propolis on viability/proliferation, migration, differentiation and cytokine expression in DPSCs.

**Methods:** Commercially available DPSCs (Lonza) were treated with aqueous extract of propolis (AEP) or ethanolic extract of propolis (EEP), and viability/proliferation was evaluated using 3-(4,5-dimethylthiazol-2-yl)-2,5-diphenyl tetrazolium bromide (MTT) assays and quantification of nuclear staining. DPSC differentiation into mineralizing cells was evaluated with Alizarin red staining and cell migration was assessed using Boyden Chamber Transwell inserts. Cytokine expression was measured by RT-qPCR. AEP and EEP at 0.03 and 0.1 mg/mL did not affect DPSC viability/proliferation for up to 7-days treatment.

**Results:** Higher doses (0.33–33 mg/mL) induced a dose dependent decrease in DPSC viability/proliferation with a more prominent effect with EEP at 7 days. Neither AEP nor EEP induced DPSC differentiation into mineralizing cells, but both AEP and EEP (0.03–0.1 mg/ml) induced a dose dependent increase in DPSC migration. In addition, EEP prevents the upregulation of IL1b and IL6 but not IL8 and CCL2 in response to lipopolysaccharide stimulation. AEP has less potent anti-inflammatory effects and prevents only IL1b upregulation.

**Conclusion:** This study provides new information about the biologic properties of ethanolic and aqueous extracts of propolis and shows that propolis, at doses that do not affect cell viability, induces DPSC migration and has anti-inflammatory properties. These data highlight the potential use of propolis as an alternative intra-canal medicament for regenerative endodontic procedures.

Corresponding author
Benoit Michot, bemichot@yahoo.fr

## INTRODUCTION

Regenerative endodontics is the study of the biological process of regenerating or engineering human dental tissues within and around the root canals to restore its normal function (*Murray, Garcia-Godoy & Hargreaves, 2007*). One of the ultimate goals of regenerative endodontic treatment is to restore the functional integrity of the pulp-dentin complex (*Kim, Kahler & Lin, 2016*). Currently, there are two different approaches of regenerating the dental pulp: the cell-based approach consisting of transplanting exogenous stem cells loaded onto scaffolds and the cell homing approach aiming to recruit endogenous stem cells into tissues to be regenerated. The cell-based approach was recently applied to humans with irreversible pulpitis where dental pulp stem cells (DPSCs) were successfully transplanted in pulpectomized teeth, proving that pulp regeneration can safely happen in human subjects (*Nakashima et al., 2017*). However, the cell-based approach faces numerous challenges, such as successfully handling the autologous stem cells, having good manufacturing practice facilities, and overcoming government regulatory policies, making the cell homing approach more clinically translatable (*Kim, Kahler & Lin, 2016*).

A critical step in a regenerative endodontic procedure is the disinfection of the canal by placing an interappointment medication with either triple antibiotic paste (TAP) or calcium hydroxide. Even though TAP has historically been the intracanal medicament of choice, recent studies have shown some limitations: TAP irreversibly stains teeth, reduces the survival of human stem cells of the apical papilla, lowers the release of bioactive molecules from dentin, and induces an inflammatory response (*Ferreira, Puppin-Rontani & Pascon, 2020*; *Pereira et al., 2014*; *Ruparel et al., 2012*). Due to these limitations, clinicians are seeking for better intracanal medications that allow the disinfection of the root canal while promoting the regeneration of the pulp-dentin complex.

Propolis is a natural substance produced by honeybees, containing about 55% resinous compounds, 30% beeswax, 10% ethereal and aromatic oils, and 5% bee pollen (*Burdock, 1998*). Propolis and its derivatives, such as phenolic and flavonoid compounds have various biological properties including antimicrobial, anti-inflammatory, antioxidant, anesthetic and cytotoxic effects (*Dantas Silva et al., 2017*; *Wagh, 2013*). Specifically, previous studies showed that propolis' antimicrobial effects target a broad spectrum of bacteria including *E. coli*, *S. aureus*, *E. faecalis* and some multidrug-resistant bacteria (*Dantas Silva et al., 2017*; *Inui et al., 2014*; *Przybyłek & Karpiński, 2019*; *Schmidt et al., 2014*). Due to the diversity of its biological properties, several studies have demonstrated the ability of propolis to be applied in the field of medicine and dentistry. In dentistry, propolis has been used for the treatment of periodontitis, root canal disinfection, and pulp capping without having any recorded allergic reactions (*Carvalho et al., 2019*; *El-Tayeb et al., 2019*; *Nakao et al., 2020*; *Abdel Hafez & Soliman, 2019*).

In addition to the adequate disinfection, the presence, proliferation and differentiation of DPSCs are fundamental to the development of the pulp-dentin complex and the success of regenerative endodontics (*Huang & Garcia-Godoy, 2014*). This emphasizes the need to understand the effect of intracanal medications on dental pulp stem cells. Previous studies

evaluating the effects of ethanolic extracts of propolis on cell migration showed that propolis induces adipocyte and chondrocyte migration but do not affects bone marrow mesenchymal stem cells (BMSC) or osteocyte migration capabilities (*Elkhenany, El-Badri & Dhar, 2019*). Further, propolis was shown to induce BMSC differentiation into osteocytes and chondrocytes, and contribute to tubular dentin formation *in vivo* (*Elkhenany, El-Badri & Dhar, 2019*; *Ahangari et al., 2012*). A recent study also showed that propolis induces differentiation of stem cells from human exfoliated deciduous teeth into mineralizing cells (*Kale et al., 2022*). However, while propolis in association with mineral trioxide aggregate (MTA) induces DPSC viability and differentiation into odontoblast-like cell, the specific effects of propolis itself on DPSC migration, differentiation, and cytokine expression is still not well understood (*Kim et al., 2019*; *Esmaeilzadeh et al., 2022*; *Nazemi Salman et al., 2023*). Thus, this *in vitro* investigation aimed to compare the effect of ethanolic and aqueous extracts of Chinese propolis on DPSC viability/proliferation, migration, differentiation into mineralizing cells and cytokine/chemokine expression.

## MATERIALS AND METHODS

### Preparation of propolis extracts

Commercially available propolis (Stakich Bee Propolis Chunks-Pure, Natural, Amazon), collected from bee farms in the Zhejiang Province of mainland China, was crushed with a mortar to obtain a fine powder and extraction was performed as described by *Khoshnevisan et al. (2019)* with some modifications for the aqueous extract of propolis (AEP) extracts. Fifty milliliters of either Millipore water or ethanol 70% was added to 5 g of propolis powder and incubated for 4 h at 37 °C with agitation. The solution was centrifuged for 10 min at $800\times g$. The supernatant of AEP was filter-sterilized (polyethersulfone membrane with 0.22 μm pores), aliquoted and stored at −80 °C. The supernatant of ethanolic extract of propolis (EEP) was collected and dehydrated in vacuum desiccator for 24 h to remove ethanol. The dried EEP collected after complete evaporation of the solvent (70% ethanol in water) was re-dissolved with 50 mL of water under agitation for 2 h at 37 °C. Then, the EEP extract was filter-sterilized (Polyethersulfone membrane with 0.22 μm pores), aliquoted and stored at −80 °C. Before use, the propolis stock solutions (100 mg/mL) were warmed up to 37 °C to dissolve any precipitate and diluted in culture media to obtain concentrations from 33 to 0.03 mg of propolis/mL (*Elkhenany, El-Badri & Dhar, 2019*; *Esmaeilzadeh et al., 2022*). For experiments testing the effects of propolis solvent, the solvent of each propolis extract was prepared as described above but without propolis powder.

### Cell culture

Commercially available human DPSCs (ref: PT-5025; Lonza, Walkersville, MD, USA) were grown in α-Minimum Essential Medium (α-MEM, Sigma-Aldrich, St. Louis, MO, United States) supplemented with 10% fetal bovine serum, L-glutamine (2 mM), ascorbic acid (100 mM), penicillin (100 U/mL), streptomycin (100 mg/mL), and amphotericin B (2.5 mg/mL). Cells were maintained at 37 °C/5% $CO_2$ and used in experiments at passage 2 to

4. DPSCs were plated at a density of 5,000 cells/cm$^2$, and culture medium was changed every 3–4 days (*Michot, Casey & Gibbs, 2020*).

## 3-(4,5-dimethylthiazol-2-yl)-2,5-diphenyltetrazolium bromide (MTT) Assay

DPSCs were plated in 96-well plates and treated with either control media, AEP or EEP at different concentrations for 3, 4 or 7 days. After treatment, DPSCs were incubated with MTT (0.5 mg/mL) for 2 h at 37 °C. The MTT metabolic product formazan was solubilized with HCl 40 mM in isopropanol for 30 min with agitation. Absorbance was measured at 570 nm using a plate reader (*Michot, Casey & Gibbs, 2020*).

## DAPI staining

DPSC proliferation was evaluated by quantifying the number of cells nuclei, with DAPI staining, after treatment with control media or propolis extracts. DPSCs were plated in 12-well plates and treated with control media, AEP or EEP (0.03–33 mg/mL) for 7 days. After incubation, the cells were washed with phosphate-buffered saline (PBS) and fixed with formalin 4% for 15 min. Non-specific staining was prevented by 1 h incubation in a blocking solution (1% triton + 1% bovine serum albumin in PBS). Cells nuclei were stained with DAPI (1 µM in PBS) for 15 min at room temperature. The cells were washed with PBS three times for 1 min before visualization with a fluorescent microscope (Revolve, Echo). From each cell culture well, five images were collected and the quantification of the number of stained nuclei was performed by a blinded experimenter, using the analyze particle tool of the Image J Software.

## Migration assay

Control media, propolis extracts (0.03, 0.1 or 3.3 mg/mL) and TAP (obtained from the Harvard School of Dental Medicine clinic; 10 mg/mL clindamycin, 10 mg/mL metronidazole and 10 mg/mL ciprofloxacin; 30 µL TAP in 0.7 mL media media) were placed in wells of a 12-well plate. DPSCs (10,000 cells/cm$^2$) were seeded on Boyden Chamber Transwell inserts containing a permeable membrane with 8.0 µm pores allowing the migration of cells from the top to the bottom surface of the membrane (MCEP12H48; Millipore). After 2 h resting to allow cell adhesion to the membrane, the transwells with the cells were placed in the 12-well plate containing propolis/TAP/control media and were incubated for 2 days at 37 °C/5% CO$_2$. The cells on the top face of the membranes, that did not migrate, were removed with a cotton swab and then the cells that migrated on the bottom face of the membrane were fixed with formalin 10% for 15 min. The cells were then stained with crystal violet (1% in water) for 20 min at room temperature. After thorough washing with millipore water, cells were visualized with light microscope (Revolve, Echo), and four images were collected from each transwell. The number of cells was manually counted by a blinded experimenter (*Takeshita-Umehara et al., 2023*).

## Alizarin red staining

DPSCs were treated with either control media, odontogenic differentiation media (ODM; β-glycerophosphate 5 mM + dexamethasone 100 nM in supplemented αMEM), AEP

(0.1 mg/mL) or EEP (0.1 mg/mL). After 21 days treatment DPSCs were fixed with 10% formalin in PBS for 15 min. DPSCs were washed with PBS, incubated with alizarin red (2% in water, pH 4.1, 15 min) and washed with deionized water before microscopic visualization of calcium nodule formation. Then, the stain was extracted by 16 h incubation with 5% isopropanol + 10% acetic acid in water and absorbance was measured at 405 nm (*Michot, Casey & Gibbs, 2020*; *Baldión, Velandia-Romero & Castellanos, 2018*).

## RNA extraction and quantitative Polymerase Chain Reaction (qPCR)

DPSCs were treated for 14 days with control media, ODM, AEP (0.1 mg/mL) or EEP (0.1 mg/mL) for differentiation assays, or 2 days with control media, lipopolysaccharide (LPS; 0.1 µg/mL), LPS + AEP (0.1 mg/mL) or LPS + EEP (0.1 mg/mL) for the quantification of cytokine expression. Total RNA was extracted using RNAzol per manufacturer's instructions, and the quality and quantity were measured using a Nanodrop. cDNA synthesis from 1 µg of total RNA and elimination of genomic DNA were performed using RT2 First Strand Kit (Qiagen). The PCR amplification was performed using 10 ng of cDNA, TaqMan Fast Advanced Master Mix (Applied Biosystem) and Assays-on-Demand Gene Expression probes (final concentration of 900 nM for primers and 250 nM for the probe; Applied Biosystems) for the following target genes: Dentin Sialophosphoprotein (DSPP, Hs00171962_m1), Dentin matrix protein (DMP1, Hs01009391_g1), Matrix Extracellular Phosphoglycoprotein (MEPE, Hs00220237_m1), alkaline phosphatase (ALP, Hs03046558_s1), Interleukin 1 beta (IL1b, Hs01555410_m1), Interleukin 6 (IL6, Hs00174131_m1), Interleukin 8 (IL8, Hs00174103_m1) and chemokine (C-C motif) ligand 2 (CCL2, Hs00234140_m1). The PCR reaction, in triplicate for each sample, was performed with a StepOnePlus Real-Time PCR System (Applied Biosciences) for 40 amplification cycles as follows: 15 s denaturation at 95 °C and 60 s annealing/elongation at 60 °C. Specific RNA levels were calculated using the ΔΔCT method (*Livak & Schmittgen, 2001*) and normalized to the housekeeping gene glyceraldehyde 3-phosphate dehydrogenase (GAPDH; Hs02758991_g1). The efficiency of the primers was evaluated as previously described by *Dankai, Pongpom & Vanittanakom (2015)* (Fig. S1). The validity of the results was checked with appropriate controls including reverse transcription control and negative PCR control (*Michot, Casey & Gibbs, 2020*).

## Data analysis

Statistical analyses were performed and figures were generated using GraphPad Prism 5 Software. Data are expressed as the mean and standard error of the mean (SEM) (*Lee, In & Lee, 2015*). The homogeneity of variances was tested with the Brown–Forsythe test. The effects of AEP and EEP in the MTT assay were analyzed with two-way ANOVA followed by the Bonferroni's *post-hoc* test. All other data were analyzed using one-way ANOVA followed by the Dunnett's *post-hoc* test for comparison to the control group or Tukey's test for multiple comparisons. Kruskal-Wallis test followed by Dunn's test was used when normality test failed. Significance was set at $p < 0.05$.

## RESULTS

### Effects of propolis extracts on DPSC viability/proliferation

DPSC viability/proliferation was evaluated using the MTT assay and it was first determined that the majority of the solvent dilutions of both AEP and EEP did not affects cell viability. Only the highest concentration of solvents (1/3 dilution, corresponding to the solvent dilution contained into the propolis extract solution at 33 mg/mL) slightly decreased DPSC viability compared to control group (Fig. S2). Aqueous extract of propolis (AEP) induced a dose-dependent decrease in DPSC viability/proliferation compared to the control group. A significant decrease in DPSC viability/proliferation was observed after 4 days treatment with AEP doses higher than 10 mg/mL ($p = 0.008$ and $p < 0.001$ for AEP 10 and 33 mg/mL respectively; Fig. 1A). Longer treatment times further affected DPSC viability, as the AEP doses higher than 3.3 mg/mL decreased DPSC viability/proliferation. AEP at doses lower than 1 mg/mL did not change DPSC viability/proliferation for up to 7 days treatment (Fig. 1A).

DPSC proliferation was also evaluated by quantifying cell numbers using DAPI to stain cell nuclei. Seven days treatment with the four lowest doses of AEP (0.03–1 mg/mL) did not affect the number of cells compared to the control group ($p > 0.95$; Figs. 1B, 1C) indicating a similar cell proliferation in these experimental groups. However, the number of DPSC treated with the highest doses of AEP is decreased compared to the control group (cell number: 125+/−10; 78+/−11; 61+/−10; 25+/−2 for control, AEP 3.3, 10, and 33 mg/mL respectively; $p < 0.01$; Fig. 1B).

Similar results were obtained in DPSCs treated with EEP. Both MTT and DAPI staining data show that the EEP doses from 3.3 to 33 mg/mL dose-dependently decreased DPSC viability/proliferation ($p < 0.05$) and the lowest EEP doses (0.03 to 1 mg/mL) did not affect viability/proliferation compared to control group (Fig. 2).

Interestingly, when comparing the effects of AEP and EEP to each other, our data shows that although both propolis extracts decreased DPSC viability/proliferation, the AEP was less cytotoxic than EEP (Fig. 3). This difference was more marked with quantification of cell numbers with DAPI staining, as EEP 0.33–10 mg/mL treated groups have a lower number of DPSC than AEP 0.33–10 mg/mL treated groups ($p < 0.002$; Fig. 3B).

### Effects of propolis on DPSCs migration and differentiation

DPSC migration and differentiation are two key processes for regenerative endodontics. First, the effects of propolis extracts were tested on DPSC migration using Boyden Chamber Transwell inserts. A 48 h treatment with AEP significantly increased the number of DPSC that migrated to the lower surface of the transwells, with a maximum effect observed for the 0.1 mg/mL dose (number of cells: 2.3 ± 0.5, 14.7 ± 3.0, 25.4 ± 4.2, in control, AEP 0.03 and 0.1 mg/mL treated groups respectively, $p < 0.002$, Fig. 4). Similar effects were found after treatment with EEP, although EEP 0.1 mg/mL is slightly less effective than AEP 0.1 mg/mL to induce DPSC migration (Fig. 4). In addition, the effects of TAP were evaluated on stem cell migration. TAP, one of the most used intra-canal medications, and propolis have been shown to have antibacterial effects but whether TAP

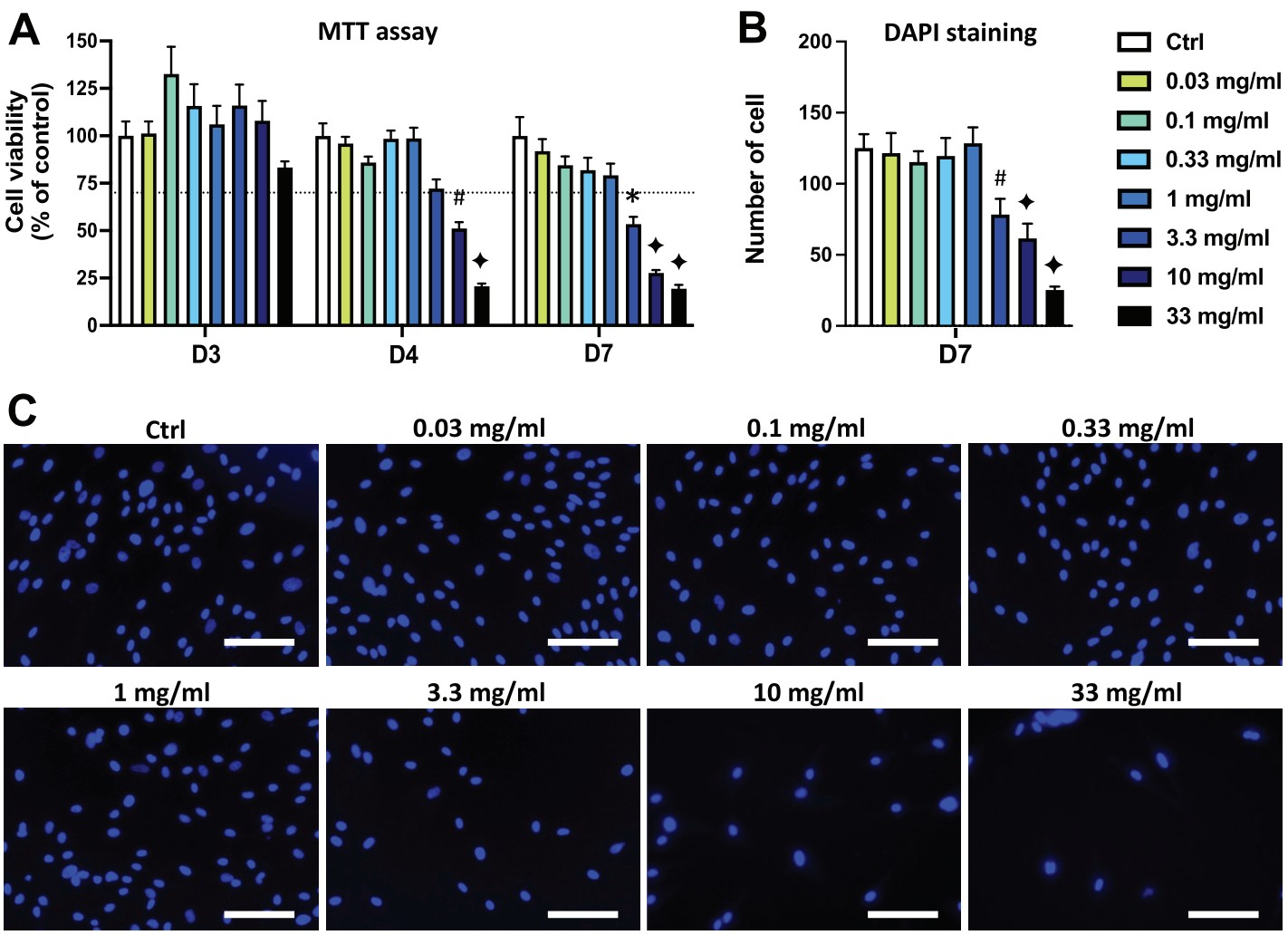

**Figure 1 Effects of aqueous extract of propolis (AEP) on DPSC viability/proliferation.** DPSCs were treated for 3, 4 or 7 days with control media or different doses of AEP (0.03–33 mg/mL) and MTT assay (A) or DAPI staining (B, C) were performed to assess cell viability and proliferation. (A, B) Bar graphs show the quantitative analysis from three independent cell cultures (18 replicates/group for MTT assay and 15 replicates/group for DAPI staining). (C) Shows representative images of DAPI staining in DPSCs treated for 7 days with AEP. Dash line indicates 30% decrease in cell viability, which represent the level of cytotoxicity. Scale bar: 130 μm. *$p < 0.05$, #$p < 0.01$, ♦$p < 0.001$ compared to control group, Dunn's multiple comparison test.

would be of potential interest for stem cell migration compared to propolis is not known. We show that TAP did not induce DPSC migration compared to control group ($p > 0.99$) suggesting propolis might have a better biological profile as an intracanal medication for regenerative endodontic procedures.

The effects of propolis on DPSC differentiation into mineralizing cells were evaluated using Alizarin red staining. DPSCs treated with odontogenic differentiation media for 21 days, which serves as positive control, shows formation of numerous calcium nodules, whereas both AEP and EEP did not induce the formation of calcium deposits in DPSCs (Figs. 5A, 5B). Similarly, gene expression analysis showed that alkaline phosphatase (ALP) was not expressed (Ct value >37.98 for each experimental groups at day 7 and 14; a CT

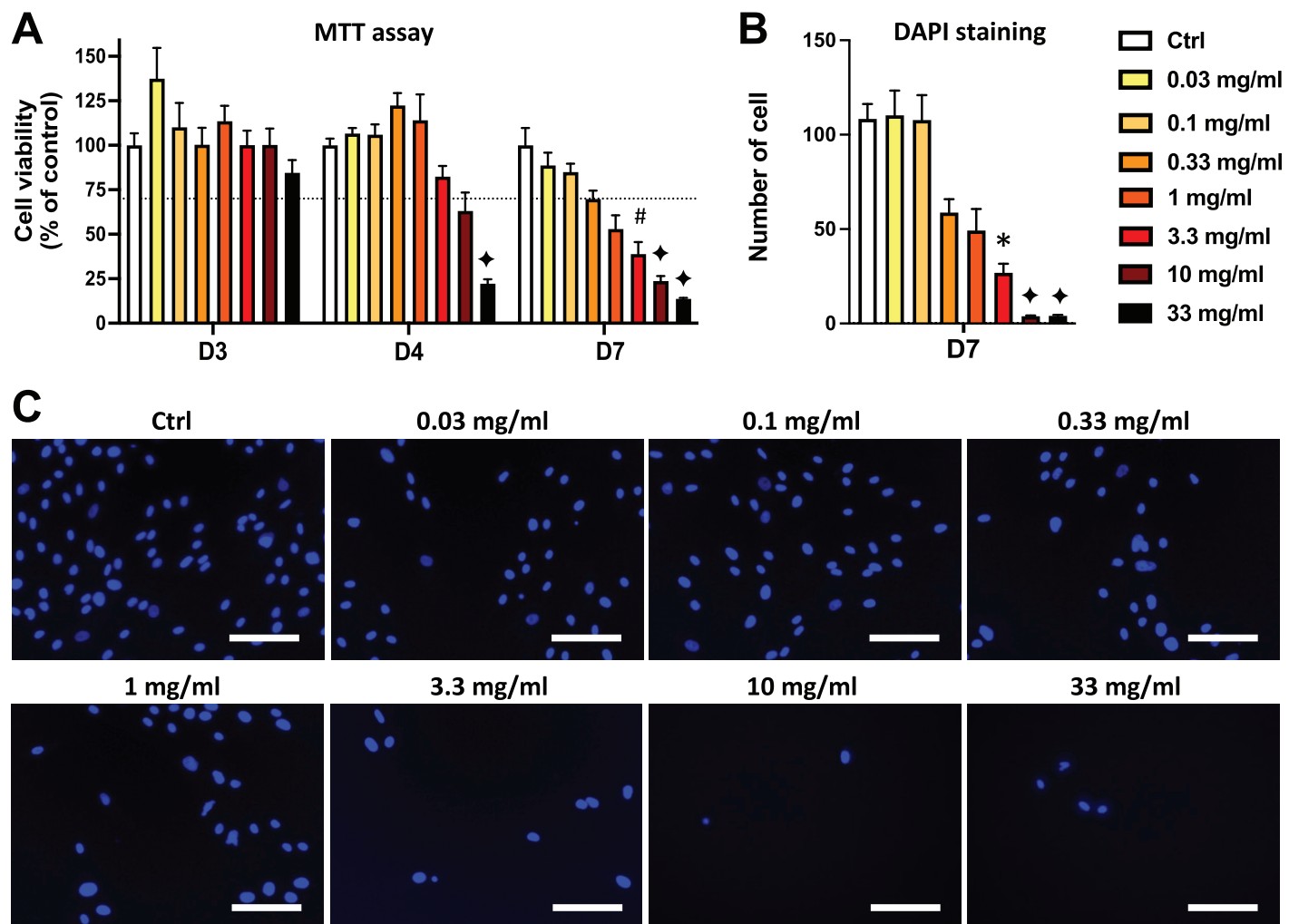

**Figure 2 Effects of ethanolic extract of propolis (EEP) on DPSC viability/proliferation.** DPSCs were treated for 3, 4 or 7 days with control media or different doses of EEP (0.03–33 mg/mL) and MTT assay (A) or DAPI staining (B, C) were performed to assess cell viability and proliferation. (A, B) Bar graphs show the quantitative analysis from three independent cell cultures (18 replicates/group for MTT assay and 15 replicates/group for DAPI staining). (C) Shows representative images of DAPI staining in DPSCs treated for 7 days with EEP. Dash line indicates 30% decrease in cell viability, which represent the level of cytotoxicity. Scale bar: 130 μm. *$p < 0.05$, #$p < 0.01$, ♦$p < 0.001$ compared to control group, Dunn's multiple comparison test.

value >35 indicates the absence of expression) and the expression of the odontoblast markers dentin matrix protein (DMP1), matrix extracellular phosphoglycoprotein (MEPE) and dentin sialophosphoprotein (DSPP) were not increased by treatment with AEP or EEP (Figs. 5C–5E), suggesting that propolis does not affect DPSC differentiation into mineralizing cells.

## Effects of propolis on pro-inflammatory cytokine expression

To determine whether propolis extracts influence DPSC inflammatory response, DPSC were stimulated for 2 days with the bacterial toxin lipopolysaccharide (LPS; 0.1 μg/mL; L3129; Sigma-Aldrich, St. Louis, MO, United States) in presence or absence of AEP (0.1 mg/mL) or EEP (0.1 mg/mL) and evaluated cytokine mRNA expression. LPS alone

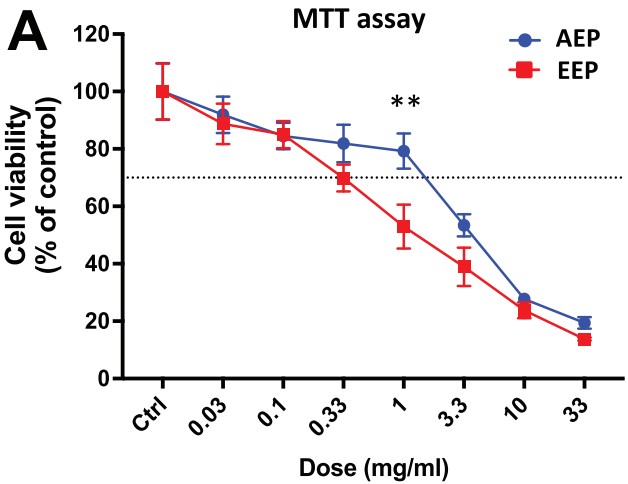

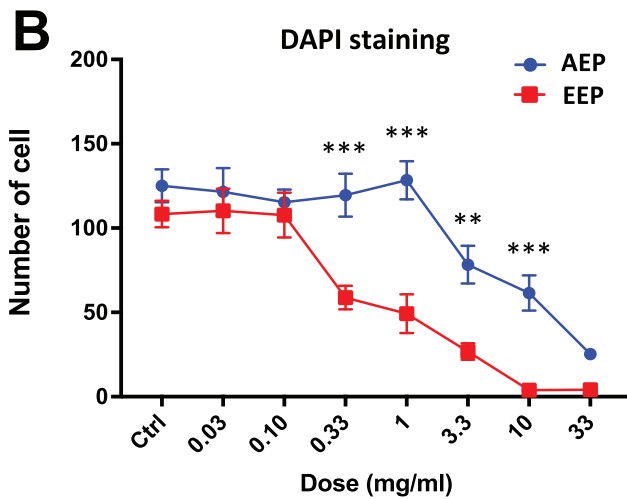

**Figure 3 Comparison of the effects of AEP and EEP on DPSC viability/proliferation.** DPSCs were treated for 7 days with either AEP or EEP (0.03–33 mg/mL) before the evaluation of viability/proliferation with MTT assay (A) or DAPI staining (B). Dash line indicates 30% decrease in cell viability, which represent the level of cytotoxicity. **$p < 0.01$, ***$p < 0.001$, AEP compared to EEP, Bonferroni's test, $n = 3$ independent cell cultures (18 replicates/group for MTT assay and 15 replicates/group for DAPI staining).             

increased the expression of IL1b, IL6, IL8 and CCL2 (fold increase: ×10, ×19, ×62, and ×37 respectively compared to control media treated group; $p < 0.001$; Fig. 6). This increase in cytokine expression was only marginally prevented by AEP as only IL1b expression was significantly reduced (9.9+/−4.0 *vs.* 1.8+/−0.4 in LPS and LPS + AEP treated group, $p = 0.03$; Fig. 6). However, EEP treatment induced a clear reduction in the expression of IL1b and IL6 (about two-fold decrease, $p < 0.05$) but no significant decrease in IL8 and CCL2 expression (Fig. 6), indicating that EEP might have a more potent anti-inflammatory effect than AEP on DPSCs.

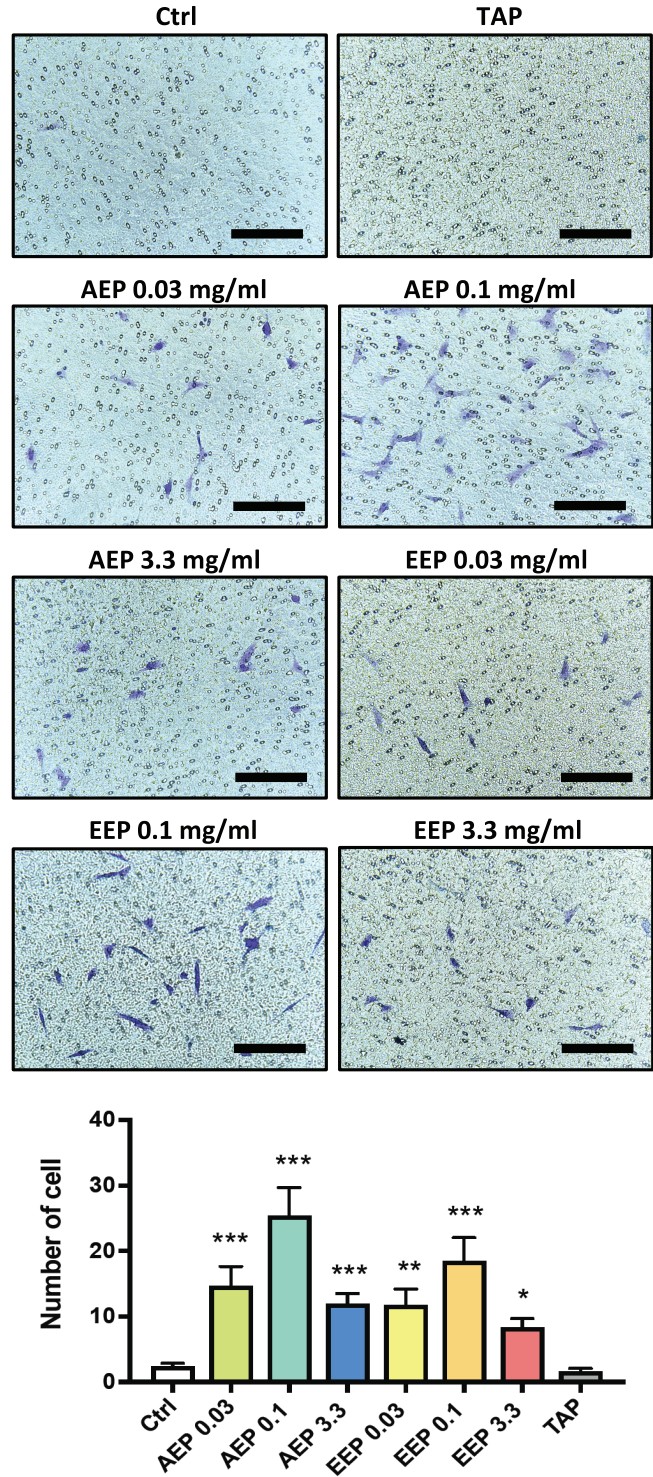

**Figure 4 Effects of AEP and EEP on DPSC migration.** DPSCs plated on Boyden Chamber Transwell inserts and exposed control media, AEP (0.03, 0.1 or 3.3 mg/mL), EEP (0.03, 0.1 or 3.3 mg/mL) or TAP for 2 days. Top panel, representative images of cells that migrated on the bottom face of transwell insert. Scale bars: 200 μm. Bottom panel, quantitative analysis of cell migration from four independent cell cultures (16 replicates/group). *$p < 0.05$, **$p < 0.01$, ***$p < 0.001$, compared to control group, Dunn's multiple comparison test.    

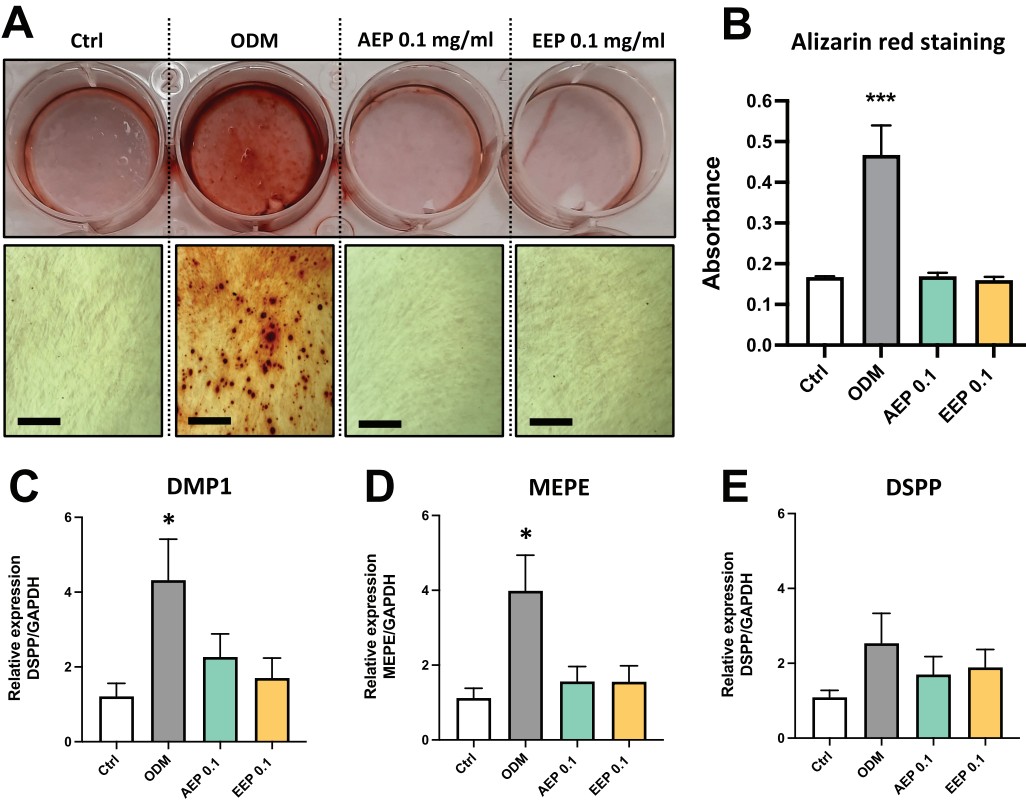

**Figure 5  Effects of AEP and EEP on DPSC differentiation.** (A) Representative images of alizarin red staining in DPSCs treated for 21 days with control media, odontogenic differentiation media (ODM), AEP or EEP. Calcium nodules are present after treatment with ODM but neither AEP nor EEP induces DPSC differentiation into mineralizing cells. (B) Quantitative analysis of alizarin staining. (C–E) mRNA expression of the odontoblast markers DMP1, MEPE and DSPP was evaluated in DPSCs treated for 14 days with control media, ODM, AEP (0.1 mg/mL) or EEP (0.1 mg/mL). Propolis extract did not affect the expression of DMP1, MEPE or DSPP. Scale bars: 200 μm. *$p < 0.05$, ***$p < 0.001$, compared to control group, Dunnett's test, $n = 3$ independent cell cultures (six replicates/group for alizarin red staining and qPCR).                                     

## DISCUSSION

This study evaluated the effects of two different Chinese propolis extracts on DPSC viability, differentiation and migration (Fig. 7). Different types of propolis are available worldwide including red Brazilian propolis, green Brazilian propolis, European Propolis and Chinese propolis which composition and properties is dependent on the plant source of the resin used by honeybees. It was reported that different types of propolis have slightly different biological properties. Previous studies showed that red propolis coming from the northeast of Brazil has better antimicrobial effects than other types of Brazilian propolis (*Machado et al., 2016*; *Koo et al., 2000*) while green propolis has lower antioxidant activities than other types of propolis (*Pazin et al., 2017*). In addition, *Zaccaria et al. (2017)* showed that brown propolis has superior activity on modulating RNA and protein expression related to oxidative stress and inflammation. This highlight that the origin of propolis is an important to criteria to consider for the evaluation of its biological activity. However, some of the chemical compounds extracted from propolis including caffeic acid phenethyl ester

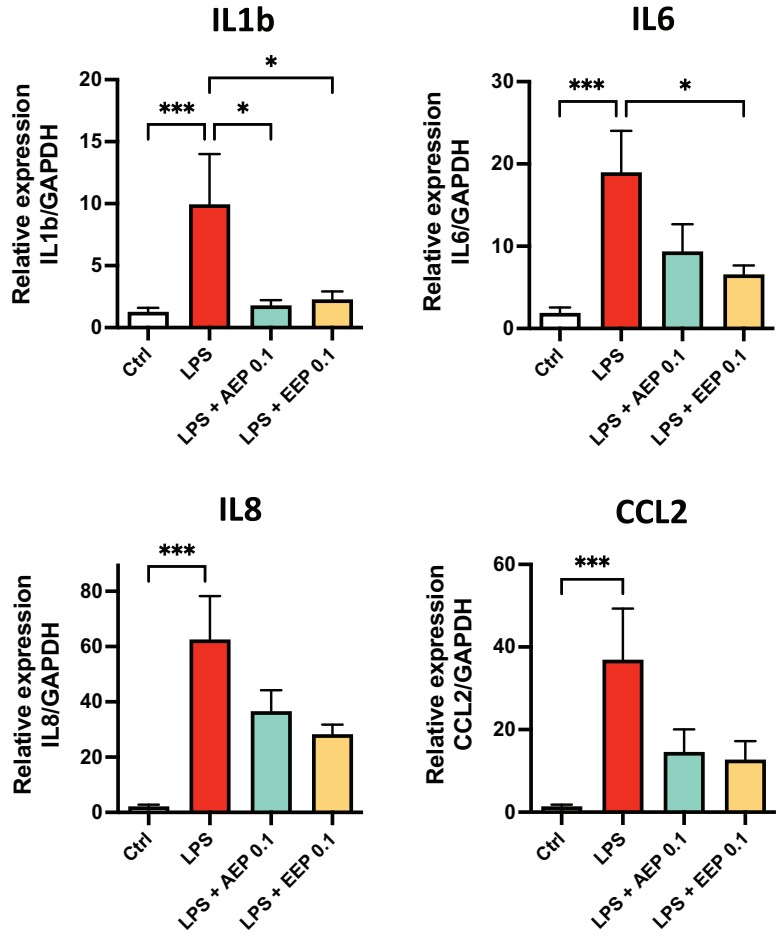

**Figure 6 Effects of AEP and EEP pro-inflammatory cytokine expression.** DPSCs were treated for 2 days with control media, lipopolysaccharide (LPS; 0.1 μg/mL), LPS + AEP (0.1 mg/mL) or LPS + EEP (0.1 mg/mL) and mRNA expression of IL1b, IL6, IL8 and CCL2 was evaluated by RT-qPCR. *$p < 0.05$, ***$p < 0.001$, Dunn's multiple comparison test, $n$ = 4 independent cell cultures (eight replicates/group).

and chrysin are of particular interest in dentistry as previous studies showed they have a strong antibacterial and anti-inflammatory properties and promote DPSC proliferation and differentiation into mineralizing cells (*Kingsley, 2022*; *Alipour et al., 2021*).

In the present study both AEP and EEP induce DPSC migration but AEP is less cytotoxic than EEP. These results are consistent with Elgendy's study that showed the cytotoxicity level of EEP was slightly higher than AEP (*Elgendy & Fayyad, 2017*). Depending on the solvent used, the chemical composition of the extract was shown to vastly change in quantity and type of molecules extracted (especially phenolic and flavonoid compounds) (*Kubiliene et al., 2015*; *Liaudanskas et al., 2021*; *Park & Ikegaki, 1998*; *Sun et al., 2015*). Overall, ethanol allows the extraction of more diverse molecules compared to AEP; for example, polyphenols such as Pinocembrin and Kaempferol are present in both EEP and AEP but isorhamnetin and Galangin are extracted only in EEP (*Kubiliene et al., 2015*; *Liaudanskas et al., 2021*; *Park & Ikegaki, 1998*; *Sun et al., 2015*). EEP

**Propolis**

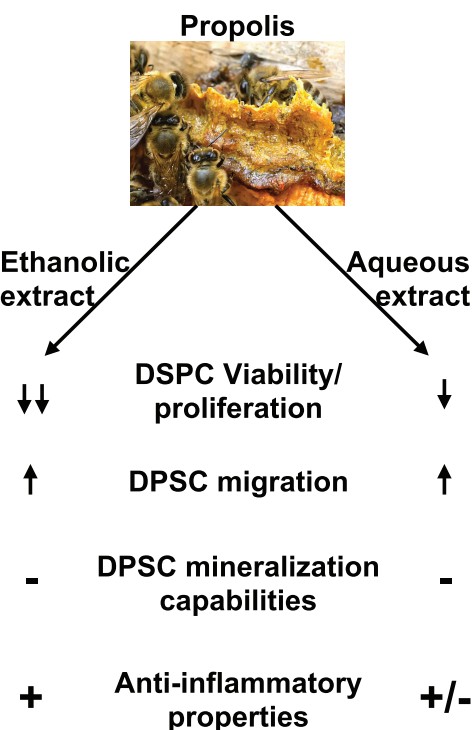

**Ethanolic extract**                                              **Aqueous extract**

↓↓        **DSPC Viability/ proliferation**        ↓

↑        **DPSC migration**        ↑

-        **DPSC mineralization capabilities**        -

+        **Anti-inflammatory properties**        +/-

**Figure 7 Summary of EEP and AEP effects on DPSC.** Both ethanolic and aqueous propolis extracts affect DPSC viability/proliferation, migration, and cytokine production but they do not induce DPSC differentiation into mineralizing cells. In addition, EEP has superior anti-inflammatory effects and decreases further DPSC viability/proliferation than AEP.

also has an overall higher content of flavonoid and polyphenol than AEP (*Kubiliene et al., 2015*; *Liaudanskas et al., 2021*; *Park & Ikegaki, 1998*; *Sun et al., 2015*). The higher quantity of polyphenols in EEP compared to AEP, might explain that, at the same dose, EEP would have increased cytotoxicity compared to AEP, especially at the highest doses of propolis used in our study. However, it is possible that some other compounds present in EEP but not AEP may mediate the increased cytotoxicity of EEP. However, further investigation is needed to determine the chemical composition of our propolis extracts and identify specific compounds that mediate the AEP and EEP effects on cell migration and cytokine expression.

The evaluation of the effects of propolis on DPSC migration showed that both EEP and AEP from 0.03 to 0.1 mg/mL led to a dose dependent increase in cell migration, but at a higher concentration, the cell migration decreased. Previous studies reported that propolis induces migration of chondrocytes and adipocytes but not osteoblasts suggesting differential sensitivity to propolis depending on cell type (*Elkhenany, El-Badri & Dhar, 2019*). Moreover, other studies investigating signaling pathways involved in propolis effects have shown that propolis increases the expression of interferon gamma and stromal cell-derived factor 1 which are key factor involved in DPSC migration (*Sá-Nunes, Faccioli & Sforcin, 2003*; *Strojny et al., 2015*). The decrease in cell migration at high doses of propolis might be the consequence of the decreased cell growth or the cytotoxicity of

propolis extracts as seen from the cell viability results. In addition, the Fig. 4 shows that TAP did not induce cell migration and previous studies showed that TAP has high cytotoxic effects against stem cells, and similar antimicrobial effects as propolis (*Ruparel et al., 2012*; *Abdel Hafez & Soliman, 2019*; *Cunha Neto et al., 2021*). Altogether, these data suggest that propolis could have a preferred biological activity for an intracanal medication than TAP for regenerative endodontic procedures.

The evaluation of the effects of propolis on odontogenic differentiation showed that both AEP and EEP did not induce DPSCs to produce mineralized substrate as shown by the absence of calcium deposits (Fig. 5A). This finding is in line with other studies which reported that bone marrow mesenchymal stromal cells showed no tendency for osteogenic differentiation in the presence of propolis (*Elkhenany, El-Badri & Dhar, 2019*). However, propolis has been shown to inhibit late stages of osteoclast maturation through reduction in the formation of actin rings which are essential for bone resorption (*Pileggi et al., 2009*). Another study showed that $TiO_2$ dental implants loaded with propolis increased the expression of collagen fibers and osteogenic differentiation (*Somsanith et al., 2018*). In addition, *Kale et al. (2022)* showed that ethanolic extracts of propolis slightly increase the formation of calcium deposits in cultured stem cells from human exfoliated deciduous teeth.

Other studies also showed that combination of propolis and MTA or calcium hydroxide enhance odontoblastic differentiation of DPSC (*Kim et al., 2019*; *Zubaidah et al., 2021*). However, propolis alone was not evaluated in these studies. In addition, *Alipour et al. (2021)* showed that chrystin, an active ingredient found in propolis, loaded in polycaprolactone/gelatin scaffold induced DPSC differentiation into mineralizing cells. The main difference between this study and others is the use of scaffold to increase solubility and bioavailability propolis extracts, which suggests that propolis loaded in scaffold would have superior effects on DPSC differentiation than propolis alone. These contradicting studies also show that there is a lack of understanding on the effects of propolis on different type of stem cells. In addition, a limitation of our study is that we performed mineralization assay with alizarin red staining *in vitro* which do not inform on the potential effects of propolis *in vivo* in pathological conditions. Further research is needed to conclude whether propolis loaded in scaffold such as hydrogels would be suitable material to enhance DPSC differentiation into odontoblasts and contribute to tertiary dentine formation *in vivo*.

Previous studies have shown that propolis extracts modulate immune reaction, edema and pro-inflammatory mediator expression in various inflammatory disorders such as ulceritis colitis arthritis and periodontitis (*Barbosa Bezerra et al., 2017*; *Borrelli et al., 2002*; *Furukawa et al., 2021*; *Zulhendri et al., 2022*). This study shows that EEP reduces cytokine expression from DPSC stimulated with LPS, but AEP show only a moderate anti-inflammatory effect.

On the contrary, common intracanal such TAP and calcium hydroxide, which have good antibacterial properties, have some limitation regarding inflammation and tissue healing. Previous studies showed both TAP and calcium hydroxide increase the expression of the potent proinflammatory cytokine IL1b and TNFa, support the persistence of

inflammation and delay pulp tissue healing *in vivo* (*Pereira et al., 2014*; *Najla & Alsalleeh, 2022*). Taken together, these data suggest that the use of propolis for root canal treatment might more relevant than common intracanal medication to reduce inflammation and enhance tissue healing.

However, this study evaluating propolis effects on inflammation are limited to a single cell type, DPSC, cultured *in vitro* and on the expression of a small number of inflammatory markers that may not reflect the complexity of inflammatory diseases that can occur *in vivo*. Indeed, previous studies have shown that AEP reduces immune cell infiltration and pro-inflammatory markers expression induced in models of pulmonary and intestinal inflammation (*Machado et al., 2012*; *Khayyal, Abdel-Naby & El-Ghazaly, 2019*). In addition, caffeic acid phenethyl ester, a chemical component of propolis, inhibit TLR4 signaling and NFkb (*Oršolić & Jazvinšćak Jembrek, 2022*; *Pahlavani et al., 2020*), suggesting that the use of both EEP and AEP would be relevant therapeutic strategies to modulate inflammatory processes induced by bacterial infection.

## CONCLUSIONS

In conclusion, we demonstrated that propolis does not affect cell viability up to doses of 0.1 mg/mL, induces DPSC migration and reduces pro-inflammatory cytokine expression. This highlights the potential use of propolis for regenerative endodontic procedures using cell homing strategies.

### Funding
This research was supported by a grant from the Krakow Harvard/Forsyth Endodontic Research Fund (HBP). The funders had no role in study design, data collection and analysis, decision to publish, or preparation of the manuscript.

### Grant Disclosures
The following grant information was disclosed by the authors:
Krakow Harvard/Forsyth Endodontic Research Fund (HBP).

### Competing Interests
The authors declare that they have no competing interests.

### Author Contributions
- Ha Bin Park conceived and designed the experiments, performed the experiments, analyzed the data, prepared figures and/or tables, authored or reviewed drafts of the article, and approved the final draft.
- Yen Dinh performed the experiments, prepared figures and/or tables, authored or reviewed drafts of the article, and approved the final draft.
- Pilar Yesares Rubi performed the experiments, authored or reviewed drafts of the article, and approved the final draft.
- Jennifer L Gibbs conceived and designed the experiments, authored or reviewed drafts of the article, and approved the final draft.
- Benoit Michot conceived and designed the experiments, performed the experiments, analyzed the data, prepared figures and/or tables, authored or reviewed drafts of the article, and approved the final draft.

## Data Availability

The raw measurements are available in the Supplemental File.

## Supplemental Information

Supplemental information for this article can be found online at http://dx.doi.org/10.7717/peerj.18742#supplemental-information.

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
