# Peer review of "Effects of aqueous and ethanolic extracts of Chinese propolis on dental pulp stem cell viability, migration and cytokine expression"

_PeerJ, doi:10.7717/peerj.18742_

## Round 0.1 · original submission · Major Revisions

Dear authors,

Based on the review reports provided by two reviewers, the manuscript demonstrates significant relevance to the field and is generally well-written. However, there are essential revisions needed, particularly in the experimental design and reporting of results. Key issues include the need for compound identification through HPLC, clarification of methods used, and addressing inconsistencies in units and scientific formatting. Additionally, justifications for methodological choices like the use of DAPI for cell viability must be provided. Therefore, the manuscript requires major revisions before it can be considered for publication.

Thank you

·

Basic reporting

-The manuscript presents results that are relevant to the field of knowledge.
-The article is well-written and meets the professional standards of courtesy and expression.
-The introduction is well-written and clearly outlines the knowledge gap it aims to address.

Here some improvements:
-Ensure that scientific names are written in italics.
-Standardize the units of measurement, as both 'mL' and 'ml' are used interchangeably.
-Verify the subscripts in CO₂ and the superscripts in cm².

Experimental design

1.-It is essential to perform High-Performance Liquid Chromatography (HPLC) or another suitable method to determine the compounds present in both extracts. Although there are no significant differences between the two extracts, the manuscript indicates that the compounds in propolis vary depending on the plant source of the resin. Therefore, it is crucial to identify the compounds present in the extracts, as these are responsible for the observed properties.

2.- Justify the use of DAPI for measuring cell viability, as it is not cited in the manuscript. DAPI is not an appropriate method for assessing cell viability; MTT is sufficient, as DAPI only stains nuclei without providing further evaluation. Alternatively, using the same ImageJ software, changes in nuclear morphology can be observed, which may suggest types of cell death (doi: 10.1186/1746-1596-9-92). If this approach is not suitable, please consider removing this section.

3.- I suggest that the cell viability graphs be presented as percentages of cell viability rather than absorbance. This approach will facilitate a clearer understanding of the impact of propolis on the cells. Additionally, presenting the data in percentage format will allow for the assessment of cytotoxic effects, as ISO 10993-5 specifies that a reduction in viability of more than 30% is considered cytotoxic.

4.- In the PCR section, it is necessary to include the concentrations at which the primers and probes were used.

5.- In lines 149 and 153, describe the type of membrane used for filtration, including the material and pore size.

6.- The analysis of variance (ANOVA) is not justified as there is no mention of whether a test for homogeneity of variances was conducted. Please include this analysis.

Validity of the findings

1. It is essential to have information on the compounds present in both extracts in order to facilitate a more thorough discussion of the results.

2.- Are the amplification efficiencies of all primers compatible with the reference gene ensured? The results of this analysis are not presented, and it is necessary for the validity of the gene expression results (Pfaffl, 2001; Pfaffl et al., 2002; Liu and Saint, 2002a; Liu and Saint, 2002b; Soong et al., 2000; Wilhelm et al., 2003).

Reviewer 2 ·

Basic reporting

Dear editor, This is an in vitro study about the “Effects of aqueous and ethanolic extracts of chinese propolis on dental pulp stem cell viability, migration and cytokine expression”. Here are some comments and questions that should be addressed trying to improve the article.

General comments
1. Please, use only MeSH terms as keywords.
2. Avoid using the terms "Our", "We" (lines 66, 73, 126, 338, 348, 414) throughout the text, prefer impersonal terms.
3. Scientific names for bacteria and microorganisms must be formatted in italics.

Experimental design

Material and methods:
1. The author declared on lines 146 and 147 “..extraction was performed as described by Khoshnevisan et al. [24] with some modifications”, why did these modifications need to be made? I suggest including a justification for these changes to clarify the reader.
2. Line 150: “…aliquoted and store at -80°C.” Please, adequate the verb “store” to the past form.
3. Line 155: “The solvent of each propolis extract was prepared as described above but without propolis powder”. I did not understand this paragraph, I suggest to include a preparation of the solvent separately of extract. For the reader it seems confuse to understand what you are calling solvent and stock solutions.
4. Line 232: “Specific RNA levels were calculated using the ΔΔCT method”, How is this method? Is it from some reference? This part was not clear.
5. Line 240: “standard error of the mean (SEM)”, is it the same as standard deviation? If yes, I suggest changing the nomenclature for Standard deviation, instead of standard error of the mean (SEM).

Validity of the findings

Results:
1. Both figures present the word “Solvant”, incorrectly, please correct the word to Solvent.
2. Fig. S1 – I suggest reducing the figure caption by placing explanations regarding procedures directly in the text, and leaving only indications that facilitate the identification of terms as captions.

Discussion:
No comments for the Discussion section.

Additional comments

The work shows potential, the methodology seems to have been done correctly, however some parts of the methodology became confusing during writing. There were some spelling and punctuation errors that made it difficult to fully understand. An English-language proofreader is suggested.

For better understanding of the figures, the texts in captions need to be reduced, leaving pertinent explanations during the course of the chapters and not in figures.

Methodologies not previously mentioned must be based on references and standards, otherwise describe the procedure performed.


Thank you!

---

## Round 0.2 · accepted · Accept

Dear authors,

Congratulations on your submission, it is well-written and scientifically sound. The research question is well-defined, and methods are detailed enough for reproducibility. The manuscript meets the journal’s standards for publication, and no further revisions are required.

·

Basic reporting

The comments were addressed appropriately.

Experimental design

The methodology comments were properly corrected, ensuring greater clarity and allowing for replication.

Validity of the findings

The results are valid thanks to the complementary modifications made in previous sections of the article.

Reviewer 2 ·

Basic reporting

The work is presented in clear language, uses correct terms, and demonstrates relevant results to the scientific community.

Experimental design

The study is based on methodologies recognized in scientific literature and is well structured.
The results presented are relevant because they present a natural material that is easily accessible and has great potential for the development of treatments, complementing published studies on the subject.

Validity of the findings

The methodology is consistent and well defined. The study demonstrated adequate control standards and experimental procedures with good precision.